# Effects of health education intervention on menstrual hygiene knowledge and practices among the adolescent girls of Pokhara Metropolitan, Nepal

**Saraswati Ghimire**[1]*, **Nand Ram Gahatraj**[1], **Niranjan Shrestha**[1], **Smriti Manandhar**[1], **Shalik Ram Dhital**[2,3,4]

**1** School of Health and Allied Sciences, Pokhara University, Kaski, Nepal, **2** Concern Center for Rural Youth, Rukumkot, Nepal, **3** Health Promotion and Education Association Nepal, Kathmandu, Nepal, **4** Home In Place, Newcastle, NSW, Australia

* sarahghimire2051@gmail.com

**Data Availability Statement:** All relevant data are within the manuscript and in other file

## Abstract

### Background

Poor menstrual hygiene practices are one of the major public health problems in Nepal. Due to persistent taboos and socio-cultural constraints, adolescent girls are often unaware of scientific facts, knowledge, and practices related to menstrual hygiene. This paper aims to assess the effects of health education intervention on menstrual hygiene knowledge and practices among adolescent girls in Pokhara Metropolitan, Nepal.

### Materials and methods

A true experimental study was conducted in two government basic schools in Pokhara Metropolitan, Nepal. The study population was adolescent girls who had attended the menarche. Firstly, a pretest with the help of a self-administered questionnaire was done to find out the socio-demographic information, knowledge and practices of menstrual hygiene. Next, health education sessions were conducted among the intervention group. Finally, after one month of intervention, a post-test was conducted among the intervention and non-intervention group. Data analysis was done through descriptive and inferential statistics.

### Results

The findings showed significant improvement in the knowledge and practice level of adolescent girls on menstrual hygiene after health education intervention. Participants in the intervention group showed a significant increase in knowledge scores from 10.0% to 67.0%, while the non-intervention group remained unchanged at 7.5%. Good menstrual hygiene practices scores in the intervention group increased significantly from 22.5% to 67.0%, whereas the non-intervention group saw a slight rise from 20.0% to 22.5%. Regarding observed practice scores in menstrual hygiene, significant improvement was observed in

**Funding:** This study was funded by Pokhara Metropolitan City, Health Division, Pokhara. The grant number for this study was 438 with NPR 40,000 (USD 300),it was utilized only for data collection due to its small amount. The funders had no role in study design, data collection and analysis, decision to publish, or preparation of the manuscript.

**Competing interests:** The authors have declared that no competing interests exist.

the intervention group (45.0% to 100.0%) in contrast to the non-intervention group (25.0% to 27.5%).

## Conclusions

This study highlights the crucial role of school health education interventions in promoting menstrual hygiene among adolescent girls. It emphasizes the importance of comprehensive educational programs tailored to early teenage girls, addressing timing, content, and delivery methods alongside ensuring the availability of Water, Sanitation and Hygiene (WASH) facilities.

## Introduction

Effective management of menstrual hygiene is a vital aspect of hygiene for women and adolescent girls from menarche to menopause [1]. This includes access to clean absorbents, facilities to change, clean, or discard these items as well as access to soap and clean water for washing the body and used absorbents [2]. Despite its importance, adolescent girls lack knowledge about menstruation and the safety measures to manage it [3].

Nearly a quarter (approximately 6.4 million) of Nepal's population consists of adolescents and an estimated 290,000 women and adolescent girls in Nepal menstruate daily [4–6]. Managing menstrual hygiene in Nepal is a significant social and health concern. Although Hinduism is the predominant religion in Nepal, the country is officially secular. Many Hindu communities regard menstruating women and girls as unclean [6].

In low-income countries like Nepal, menstrual hygiene management (MHM) for school girls is a neglected issue where 15 to 22% of girls still miss school due to their menstruation-related issues [7]. Hygiene-related behavior during menstruation is vital for reproductive health and well-being [8]. Adolescence is a transition period from childhood to adulthood where changes in behavior, attitudes, moral principles and intellectual capacity occur. Healthy growth throughout this phase could improve quality of life and health [9]. Parents, teachers and peers have a significant role in improving menstrual hygiene practices [10]. However, the majority of school-going adolescent girls are unaware of the basic facts of menstruation [5].

Even though the Government of Nepal had allocated a budget of USD 15.2 million for the free distribution of sanitary pads in public schools in the year 2020–2021 and updated the national school curriculum on menstrual hygiene in textbooks from grades 4 to 12 [11] still more than half adolescents have inadequate information about menstruation and only one in ten are practicing good menstrual hygiene where high level of restriction (like dietary limitations, complete separation) is followed [12].

Due to persistent taboos and socio-cultural constraints, teenage girls are often unaware of the scientific facts and sanitary health practices related to menstruation, which have a negative impact on their health [13,14]. Studies conducted in developing countries including Nepal show that the majority of school-going adolescent girls lack knowledge and have a low level of hygienic practices regarding menstrual hygiene [6,10,15,16]. Urinary Tract Infections (UTIs) and Reproductive Tract Infections (RTIs) such as bacterial vaginosis and vulvovaginal candidiasis are serious public health problems, especially in low-income countries. These problems have been linked to poor menstrual hygiene practices and problems are more common among those who are likely to use clothes, rags or damaged reusable pads compared to the use of sanitary pads [17]. Studies conducted in resource-poor settings like South Asian and Sub-Saharan

African schools showed that girls who attend public schools are more likely to have poor menstrual hygiene because of their poor socioeconomic status [18]. Therefore, it is necessary to educate girls regarding menstruation and appropriate menstrual hygienic practices using a suitable program, especially in schools [19].

Studies conducted in Nepal, Saudi Arabia, Kenya, South Africa and India have recommended conducting menstrual hygiene and health interventions as early as possible to equip adolescent girls with scientific knowledge and hygienic practices regarding menstruation [16,20–24]. The use of various health education methods, media and communication channels such as lectures, demonstrations, audio-visual aids and the provision of resources like sanitary pads, underwear and hand-washing with soap facilities have been reported to be effective in increasing knowledge and improving menstrual practice among school-going girls [25]. Many cross-sectional studies have been conducted earlier, however there are limited intervention studies to improve menstrual hygiene among school-going adolescent girls in Nepal. Therefore this paper aims to assess the effects of health education intervention on menstrual hygiene knowledge and practices among adolescent girls in Pokhara Metropolitan, Nepal.

The study findings will help adolescent girls manage menstruation better and practice good hygiene, leading to improved menstrual health. Teachers and administrators will become better equipped to support and create a more supportive school environment. Additionally, concerned authorities such as policymakers will gain insights to develop targeted and effective menstrual health curricula and policies.

## Materials and methods

### Study design and setting

A true experimental design was employed, focusing on government basic schools in Pokhara Metropolitan, Nepal. These schools were chosen because girls in public schools are more likely to have poor menstrual hygiene due to lower socioeconomic status which is supported by previous studies conducted in South Asian and Sub-Saharan African countries [18].

### Study population

Adolescent girls aged between 10 and 15 years from two government basic schools who have attained menarche were the study population. Studies conducted in low and middle-income countries such as Nepal, India, Turkey and Nigeria [5,26–29] revealed that the mean age of menarche among school-going adolescent girls falls within the range of 10 to 15 years. A study conducted by Fetohy EM in Egypt and Chang and Chen in the Hualien Region has highlighted the importance of expanding menstrual hygiene education to elementary, preparatory and other secondary schools, especially during crucial transitional periods such as menarche. Furthermore, their studies have demonstrated the effectiveness of educational initiatives in these populations in enhancing menstrual health [20,26]. Therefore, the age group of 10 to 15 years was selected as the target population for this study.

### Sample size calculation

The total sample size for intervention and non-intervention was determined using the formula from the study conducted in India [30].

$$n = \left(Z_{1-\frac{\alpha}{2}} + Z_{1-\beta}\right)^2 \frac{p1(1-p1)+p2(1-p2)}{(p1-p2)^2}$$

n- Sample size for intervention and non-intervention groups

p1- Proportion of correct responses before intervention = 0.51

p2- Proportion of correct responses after intervention = 0.82

$Z_{1-\frac{\alpha}{2}}$-Percentile of the standard normal distribution and equal to1.64 for the alpha of 0.05 (one-sided test),

$Z_{1-\beta}$ is the percentile of the standard normal distribution and equal to 1.28 for the power of 0.9

α-level of significance

Substituting values in the formula,

n = ((0.51) *(1–0.51) + (0.82)*(1–0.82))*(1.28+1.64)$^2$/ (0.51–0.82)$^2$

n = 36

With 10.0% dropout rate,

Final sample size (n) = 40 for each group.

The proportion of correct answers before and after the intervention was considered to calculate the sample size using 0.05 as the level of significance (for one-sided test) and 0.9 as a power.

The questionnaire used by Haque et al [31] is the closest to the proposed study in terms of scoring and a number of items and reported that participation in the educational program increased knowledge from 51.0% to 82.0% and overall good menstrual practices from 28.8% to 88.9%.

## Sampling techniques

Fig 1 shows the sampling procedures that was carried out in this study. Two government basic schools of Pokhara Metropolitan was selected purposively that have better feasibility of work, matching the predetermined sample size, and situated at a distance from each other so that percolation of messages could be prevented between the two schools during the intervention phase. Then two schools were randomly assigned as intervention and non-intervention. The sampling frame of all eligible adolescent girls was prepared from the attendance register maintained at each classroom from class six to eight.

## Inclusion and exclusion criteria

Adolescent girls of government basic schools who attained menarche with regular menstrual cycles and were between the ages of 10 to15 years, who were present on the day of the study, participants who gave assent and whose parents gave consent and were willing to take part in the study were included. However, adolescent girls who were unable to participate in the study due to personal causes and who had irregular periods, who were not willing to take part in the study, participants who did not give assent and parents who did not give consent for the study were excluded from this study. Participants who had irregular periods were not included because those with irregular menstrual cycles may not experience the full benefits of the intervention or may have difficulty adhering to the study's practical aspect which was done through simulation.

## Study tools

A structured questionnaire was used for data collection. Consultation with other researchers was done for further validation of the questionnaire which consists of three parts; socio-demographic information, existing knowledge about menstruation and menstrual hygienic practices during menstruation. An observation checklist was also used to assess menstrual hygiene practices.

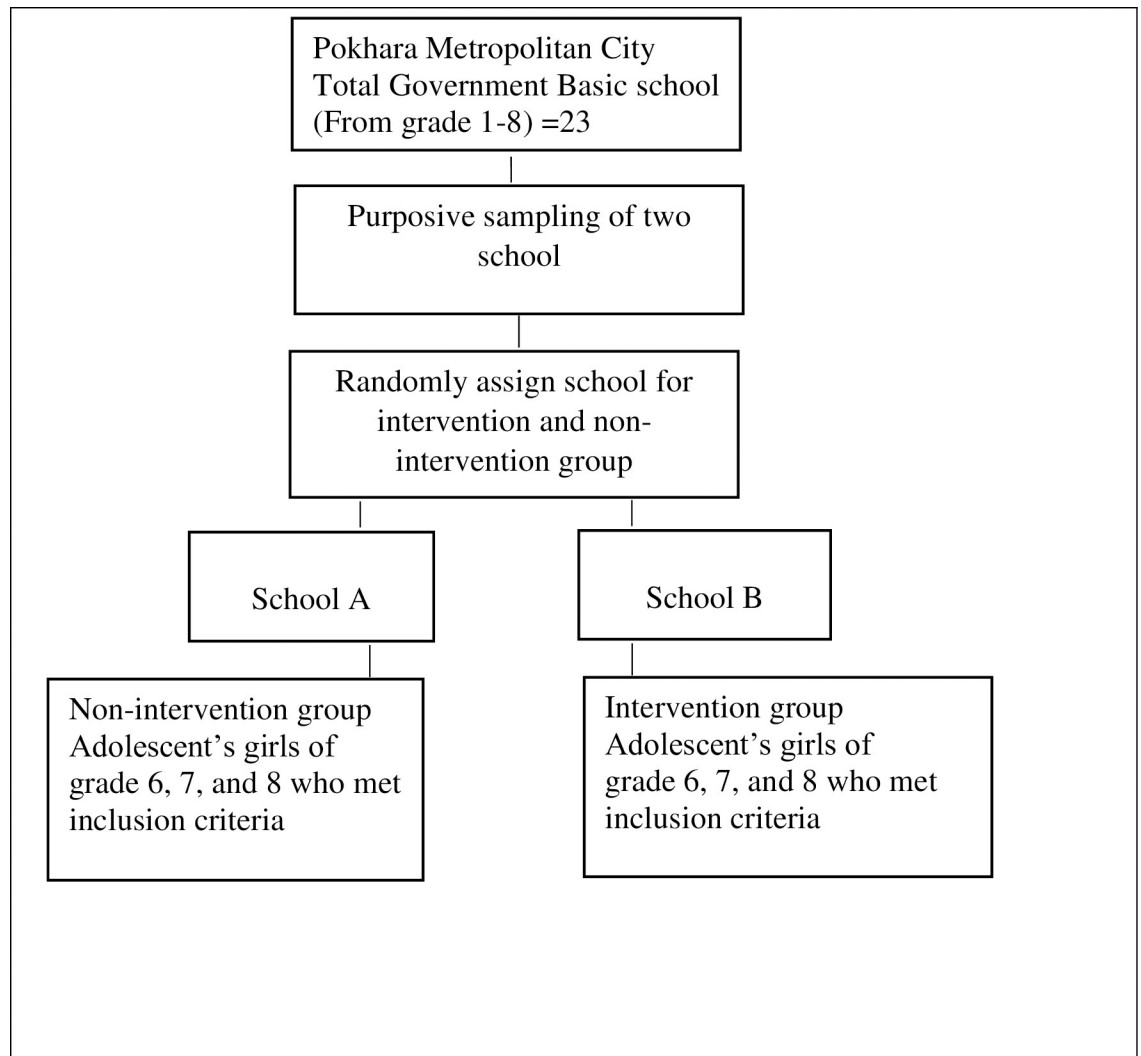

**Fig 1. Flow diagram of sampling technique process.**

### Pre-testing, reliability and validity of tools

Pre-testing of questionnaire was done in 10% of the total sample in a similar setting excluding the study area and necessary modification was made accordingly. To ensure the validity of the study, the questionnaire was developed only after the extensive literature review and was prepared under the guidance of the supervisor and senior researchers and they were also consulted for appropriate suggestions on study design, tools and methodology. Translation and back translation (English-Nepali-English) of the questionnaire were done.

### Data collection procedure

The data collection was carried out using a self-administered questionnaire. Permission was obtained from the school administration before the study. A pre-test was conducted in the intervention and non-intervention groups by using a structured questionnaire. After two days, health education sessions were provided to the intervention group through videos, mini-lectures, simulations, group discussions and handout distribution. After one month, a post-test was conducted in both groups.

| Groups | Pretest | Intervention | Posttest |
|---|---|---|---|
| Intervention | O1 | X | O2 |
| Non-intervention | O1 | No intervention | O2 |

**Keys**
O1 –Pretest of knowledge and practice regarding menstrual hygiene
X- Intervention
O2 –Posttest of Pretested knowledge and practice regarding menstrual hygiene

## Development of health education intervention package

A formative research was conducted to collect information on existing knowledge and practices related to menstrual hygiene among school-going adolescent girls. Based on the information received, the content of the intervention package was developed using various sources like National Health Education, Information and Communication Center (NHEICC), Kathmandu, Nepal, Rupantaran(transformation) training book, references from visible impact and scientific literature related to menstrual hygiene. Furthermore, ideas from menstrual hygiene management experts were taken and given through various methods and media including mini-lectures, group discussions, videos, leaflet distribution and simulation. The information was developed in such a way that it was understandable to the participants.

## Study phases

**Pre-intervention phase.** In this phase, school-going adolescent girls who met inclusion criteria and agreed to participate in the study were assessed and data was collected by using a pretested self-administered questionnaire.

**Intervention phase.** School-going adolescent girls were provided with detailed menstrual hygiene education that was intended to increase their knowledge on menstruation and encourage them for the best menstrual hygiene practices. Health education intervention was of two hours ten minute session that included interactive learning sessions using videos, PowerPoint presentations, simulation, group discussion and handouts distribution for two days. It was given only once because we intend to use resources efficiently. Secondly, the participants in repetitive educational interventions may feel pressure to conform to perceived social norms or expectations, leading to social desirability bias. They may provide responses that they believe are socially acceptable or desirable, rather than expressing their genuine opinions or behaviors because they are aware of being observed or studied. More focus was given on good hygiene habits by simulation approach such as appropriate placement of sanitary pad, use of soap during hand washing and proper disposal of used absorbents. In the simulation session, every girl was involved in the practices. At the end of the session, participants were divided into groups to discuss and dispel any myths or restrictions they may have on menstruation.

**Post-intervention phase.** The effectiveness of the health education intervention was assessed using a similar set of pre-tested questionnaires after one month of health education intervention. No sessions were conducted for the non-intervention group during the intervention. However, they received health education intervention after the post-test survey.

## Data management and analysis

The collected data were carefully handled and stored properly before and after data collection. To ensure data accuracy, each questionnaire was thoroughly reviewed every day after data

collection. For data entry, data-entry marks were created and EpiData software version 3.1 was used to minimize error. To minimize errors within the limit, 10% of the randomly selected data was manually rechecked. All the data were saved in a password-protected computer folder to ensure the security of the data and was exported from EpiData 3.1 to SPSS version 21 for further analysis as per plan. As per the need, the data were transformed and computed in SPSS 21. A descriptive analysis was done to calculate frequencies and percentages.

**Inferential statistics.** For inferential statistics, continuous data were checked for normality using Kolmogorov-Smirnov test. Data were not found to be normal so a non-parametric test Wilcoxon signed-rank test was used to assess the effectiveness of the intervention. Chi-square and Fisher's exact test was also carried out to assess whether there was a significant difference among demographic and menstrual baseline characteristics to ensure homogeneity among intervention and non-intervention groups. These results were considered significant at the 5% level. Scores were used to assess participant's knowledge and hygienic practices during menstruation. Questions were scored as follows: 1 point for a correct response, 0 point for a wrong or no answer. Self-reported knowledge and practice level were categorized based on a previous study [31,32] which is mentioned below while observed practice scores on menstrual hygiene like washing hands and appropriate disposal were categorized based on an average value which was 2.5.

Knowledge score Practice score Observed practice score through observation checklist
Poor Knowledge (0–3) Poor Practice (0-4) Poor Practice $\leq 2.5$
Medium Knowledge (4–7) Fair Practice (5-8) Good Practice ($>2.5$)
High Knowledge (8–10) Good Practice ($\geq 9$)

## Ethical consideration

Ethical approval was taken from Institutional Review Committee, Research Department, Pokhara University with reference number 185-079/080. Permission was taken from Pokhara Metropolitan Education Division and selected government basic schools to carry out the study. Assent from participants and consent from parents of adolescent was taken prior to data collection.

## Results

### Socio-demographic characteristics of the participants

The majority of participants in both intervention and non-intervention groups had ages ranging from 12 to 13 years. The median age of participants in the intervention and non-intervention groups was 13 years. Regarding education level, the majority of participants were in grade six, comprising 47.5% and 42.5% in the intervention and non-intervention groups respectively. Hindus made up 85.0% of the intervention group and 92.5% of the non-intervention group. Over half of the adolescent girls in both the intervention and non-intervention groups were Dalits, comprising 57.5% and 50.0%, respectively. Meanwhile, 70.0% of the participants in the intervention group and 67.5% in the non-intervention group were from nuclear families.

Concerning the participant's mother's educational level, more than half of the mothers were illiterate (57.5%) in intervention group while in the non-intervention group only (35.0%) of them were illiterate. The education level of participant's fathers who completed basic level education was slightly higher in the non-intervention group (40.0%) than in the intervention group (35.0%). Half (50.0%) of participant's mothers were housemaker in the intervention group and (47.5%) in the non-intervention group, the main occupation of fathers was daily wages in both groups (like Painter, Driver, Coolie); 77.5% in the intervention group and 72.5% in non-intervention group (Table 1).

**Table 1. Socio-demographic characteristics of the participants.**

| Variable | Interventional group | | Non-interventional group | |
|---|---|---|---|---|
| | (n = 40) | (%) | (n = 40) | (%) |
| **Age** | | | | |
| 10–11 years | 8 | 20.0 | 3 | 7.5 |
| 12–13 years | 18 | 45.0 | 23 | 57.5 |
| 14–15 years | 14 | 35.0 | 14 | 35.0 |
| | Median age = 13 years<br>Interquartile Range = 2 | | Median age = 13 years<br>Interquartile Range = 2 | |
| **Education Level** | | | | |
| Class 6 | 19 | 47.5 | 17 | 42.5 |
| Class 7 | 16 | 40.0 | 14 | 35.0 |
| Class 8 | 5 | 12.5 | 9 | 22.5 |
| **Religion** | | | | |
| Hindu | 34 | 85.0 | 37 | 92.5 |
| Buddhist | 4 | 10.0 | 0 | 0.0 |
| Christian | 2 | 5.0 | 3 | 7.5 |
| **Ethnicity** | | | | |
| Dalit | 23 | 57.5 | 20 | 50.0 |
| Janajati | 13 | 32.5 | 14 | 35.0 |
| Brahmin/Chhetri | 3 | 7.5 | 4 | 10.0 |
| Madhesi | 1 | 2.5 | 2 | 5.0 |
| **Family Type** | | | | |
| Nuclear | 28 | 70.0 | 27 | 67.5 |
| Joint | 12 | 30.0 | 13 | 32.5 |
| **Educational Level of Mother** | | | | |
| Illiterate | 23 | 57.5 | 14 | 35.0 |
| Non-formal education | 13 | 32.5 | 14 | 35.0 |
| Basic education | 3 | 7.5 | 11 | 27.5 |
| Secondary | 1 | 2.5 | 1 | 2.5 |
| **Educational Level of Father** | | | | |
| Illiterate | 7 | 17.5 | 8 | 20.0 |
| Non-formal education | 13 | 32.5 | 13 | 32.5 |
| Basic education | 14 | 35.0 | 16 | 40.0 |
| Secondary | 6 | 15.0 | 3 | 7.5 |
| **Occupation of Mother** | | | | |
| Business | 4 | 10.0 | 7 | 17.5 |
| Daily Wages | 16 | 40.0 | 13 | 32.5 |
| Foreign Job | 0 | 0.0 | 1 | 2.5 |
| Housemaker | 20 | 50.0 | 19 | 47.5 |
| **Occupation of Father** | | | | |
| Business | 5 | 12.5 | 8 | 20.0 |
| Service | 0 | 0.0 | 1 | 2.5 |
| Daily Wages | 31 | 77.5 | 29 | 72.5 |
| Foreign job | 4 | 10.0 | 2 | 5.0 |

## Menstrual baseline characteristics of the participants

The median age of menarche in intervention and non-intervention groups was 12 years. Participants who were aware of menstruation before menarche in the intervention group was

**Table 2. Menstrual baseline characteristics of the participants.**

| Variable | Intervention group | | Non-intervention group | |
|---|---|---|---|---|
| | (n = 40) | (%) | (n = 40) | (%) |
| **Age at menarche** | | | | |
| 10–11 years | 19 | 47.5 | 15 | 37.5 |
| 12–13 years | 20 | 50.0 | 24 | 60 |
| 14–15 years | 1 | 2.5 | 1 | 2.5 |
| Median age = 12 years, Interquartile Range = 1 | | | Median age = 12 years, Interquartile Range = 2 | |
| **Knowledge prior to menarche** | | | | |
| Yes | 27 | 67.5 | 19 | 47.5 |
| No | 13 | 32.5 | 21 | 52.5 |
| **Source of menstruation information** | | | | |
| Mother | 33 | 82.5 | 36 | 90.0 |
| Male teacher | 0 | 0.0 | 1 | 2.5 |
| Friends | 2 | 5.0 | 0 | 0.0 |
| Female teacher | 3 | 7.5 | 1 | 2.5 |
| Siblings | 2 | 5.0 | 2 | 5.0 |

67.5%, while in the non-intervention group, it was only 47.5%. The main source of menstruation information providers for the first time were mothers in both intervention (82.5%) and non-intervention groups (90.0%) (Table 2).

There was no statistically significant difference between the intervention and non-intervention groups in all of their socio-demographic and menstrual baseline characteristics at a 95 percent confidence interval (p<0.05) making a comparison possible between the two groups. Categorization was done based on previous studies conducted in Dang district and Pokhara, Nepal [33,34] (Table 3).

## Menstrual hygiene knowledge in intervention and non-intervention group

Regarding menstrual hygiene knowledge in the intervention group, only 45.0%, 32.5%, and 45.0% were aware of menstruation being a normal phenomenon, its cause and the normal age to attain menarche in the pretest. However, in the post-test, there was a statistically significant improvement (p<0.001) with responses rising to 90.0%, 85.0%, and 92.5%, respectively.

Similarly, there was also a major increase in response in the post-test regarding organ responsible for menstrual blood flow (35.0% vs 85.0%), awareness of menstrual hygiene (65.0% vs 100.0%), infection due to poor menstrual hygiene (37.5% vs 70.0%), no pregnancy during menstruation (22.5% vs 60.0%) and menstrual blood not being impure (32.5% vs 82.5). In general, there was a significant improvement in participant's self-reported high knowledge of menstrual hygiene from 10.0% to 67.5%.

In the non-intervention group, there was minimal difference between the pretest and post-test in knowledge-related items except one that heard about menstrual hygiene (67.5% vs 90.0%). Overall non-intervention group demonstrated no improvement in their high knowledge level, remaining at 7.5% (Table 4).

## Menstrual hygiene practices in intervention and non-intervention group

Regarding absorbents used by participants in the intervention group, no significant differences were observed in disposable sanitary pads (30.0% vs 27.5%) and sanitary pads with homemade reusable cloth (70.0% vs 72.5%).

**Table 3. Difference between intervention and non-intervention groups on socio-demographic and menstrual baseline characteristics.**

| Variable | Intervention group | | Non-intervention group | | p-value |
|---|---|---|---|---|---|
| | **(n = 40)** | **(%)** | **(n = 40)** | **(%)** | |
| **Age** | | | | | |
| ≤13 years | 26 | 65 | 26 | 65 | 1.000* |
| >13 years | 14 | 35 | 14 | 35 | |
| Median age = 13 years, Min = 10, Max = 15 | | | Median age = 13 years Min = 11, Max = 15 | | |
| **Education level** | | | | | |
| Class 6 | 19 | 47.5 | 17 | 42.5 | 0.500* |
| Class 7 | 16 | 40.0 | 14 | 35.0 | |
| Class 8 | 5 | 12.5 | 9 | 22.5 | |
| **Religion** | | | | | |
| Hindu | 34 | 85.0 | 37 | 92.5 | 0.481** |
| Others | 6 | 15.0 | 3 | 7.5 | |
| **Ethnicity** | | | | | |
| Brahmin/Chhetri | 3 | 7.5 | 4 | 10.0 | 0.875* |
| Janajati | 13 | 32.5 | 14 | 35.0 | |
| Dalit and others | 24 | 60.0 | 22 | 55.0 | |
| **Family type** | | | | | |
| Nuclear | 28 | 70.0 | 27 | 67.5 | 0.809* |
| Joint | 12 | 30.0 | 13 | 32.5 | |
| **Educational level mother** | | | | | |
| Illiterate | 23 | 57.5 | 14 | 35.0 | 0.079* |
| Literate | 17 | 42.5 | 26 | 65.0 | |
| **Educational level father** | | | | | |
| Illiterate | 7 | 17.5 | 8 | 20.0 | 0.775* |
| Literate | 33 | 82.5 | 32 | 80.0 | |
| **Occupation of mother** | | | | | |
| Housemaker | 20 | 50.0 | 19 | 47.5 | 0.823* |
| Other | 20 | 50.0 | 21 | 52.5 | |
| **Occupation of father** | | | | | |
| Informal Occupations | 36 | 90.0 | 37 | 92.5 | 1.000** |
| Formal Occupations | 4 | 10.0 | 3 | 7.5 | |
| **Menstrual Baseline characteristics** | | | | | |
| **Age at menarche** | | | | | |
| ≤12 years | 33 | 82.5 | 30 | 75.0 | 0.412* |
| >12 years | 7 | 17.5 | 10 | 25.0 | |
| Median = 12 years, Min = 10, Max = 14 | | | Median = 12 years, Min = 10, Max = 14 | | |
| **Knowledge prior to menarche** | | | | | |
| Yes | 27 | 67.5 | 19 | 47.5 | 0.070* |
| No | 13 | 32.5 | 21 | 52.5 | |
| **Source of menstruation information** | | | | | |
| Mother | 33 | 82.5 | 36 | 90.0 | 0.330* |
| Others | 7 | 17.5 | 4 | 10.0 | |

(*Chi-square test

** Fisher's exact test).

**Table 4.  Menstrual hygiene knowledge in intervention and non-intervention group.**

| Variable | Intervention group (n = 40) | | | | Wilcoxon signed rank test (p-value) | Non-intervention group (n = 40) | | | | Wilcoxon signed rank test (p- value) |
|---|---|---|---|---|---|---|---|---|---|---|
| | Pretest | | Posttest | | | Pretest | | Posttest | | |
| | n | % | n | % | | n | % | n | % | |
| Menstruation as a physiological process | 18 | 45.0 | 36 | 90.0 | <0.001 | 27 | 67.5 | 27 | 67.5 | 1.00 |
| Hormones as a cause of menstruation | 13 | 32.5 | 34 | 85.0 | <0.001 | 18 | 45.0 | 21 | 52.5 | 0.257 |
| Normal age to attain menarche | 18 | 45.0 | 37 | 92.5 | <0.001 | 25 | 62.5 | 26 | 65.0 | 0.705 |
| Normal duration of menstruation | 14 | 35.0 | 24 | 60.0 | 0.012 | 22 | 55.0 | 20 | 50.0 | 0.527 |
| Menstruation does not occur during pregnancy | 9 | 22.5 | 24 | 60.0 | 0.002 | 2 | 5.0 | 5 | 12.5 | 0.083 |
| Organ responsible for menstrual blood flow | 14 | 35.0 | 34 | 85.0 | <0.001 | 7 | 17.5 | 10 | 25.0 | 0.180 |
| Normal interval of the menstrual cycle | 9 | 22.5 | 24 | 60.0 | <0.001 | 14 | 35.0 | 15 | 37.5 | 0.705 |
| Heard about menstrual hygiene | 26 | 65.0 | 40 | 100.0 | <0.001 | 27 | 67.5 | 36 | 90.0 | 0.007 |
| Poor menstrual hygiene leads to infections | 15 | 37.5 | 28 | 70.0 | <0.001 | 12 | 30.0 | 12 | 30.0 | 1.000 |
| Menstrual blood is not impure | 13 | 32.5 | 33 | 82.5 | <0.001 | 7 | 17.5 | 4 | 10.0 | 0.083 |
| **Knowledge level on menstrual hygiene** | | | | | | | | | | |
| Poor knowledge (0–3) | 22 | 55.0 | - | - | <0.001 | 17 | 42.5 | 14 | 35.0 | 0.098 |
| Medium knowledge (4–7) | 14 | 35.0 | 13 | 32.5 | | 20 | 50.0 | 23 | 57.5 | |
| High knowledge (8–10) | 4 | 10.0 | 27 | 67.5 | | 3 | 7.5 | 3 | 7.5 | |

The significant differences are highlighted in bold.

In the intervention group, among users of reusable cloths, an improvement was observed in washing it with soap water, increasing from 85.7% to 100%. There was a significant improvement in the post-test in comparison to the pretest regarding sun-drying of reused cloth (21.4% vs 100.0%), changing of absorbents every 4 to 6 hours (22.5% vs 82.5%), daily bathing during menstruation (5.0% vs 62.5%), cleaning of genital every time using the toilet (50.0% vs 97.5%) and use of soap to clean genital (27.5% vs 80.0%). The significant result can also be seen in practice regarding the disposal of absorbents from the pretest to the post-test that is (57.7% vs 100.0%). Regarding restrictions, no significant progress was observed in the posttest, like temple visits and carrying out daily household chores. However, significant differences regarding limitations on certain types of food were observed.

Overall, the intervention group shows significant improvement in good menstrual hygiene practices from 22.5% to 67.5% and negligible progress can be observed in the non-intervention group from 20.0% to 22.5% (Table 5).

The observation checklist on menstrual hygiene revealed that during both the pretest and post-test observations, all participants from both groups consistently chose sanitary pads as their absorbent of choice, despite being presented with various other options such as reusable cloth pads and rags. However, in self-reported practice questions, the majority's choice was both reusable cloth and sanitary pads. It might be due to poor economic status which limits access to high-cost sanitary products. There was no hand washing practice before changing absorbents in the intervention group however after intervention this practice increased to 67.5%. On the other hand, in the non-intervention group, only 2.5% washed their hand in the pretest and none in the posttest. Slightly more than half (52.5%) of the participants were placing menstrual products following the steps in the pretest however after intervention 90% of the adolescent girls were able to place absorbents appropriately. In the non-intervention group only 40.0% were able to place sanitary pads correctly there was a slight increase in this percentage in the posttest which was 45.0%.

**Table 5. Menstrual hygiene practices in the intervention and non-intervention group.**

| Variable | Intervention group (n = 40) | | | | Wilcoxon signed rank test (p value) | Non-intervention group(n = 40) | | | | Wilcoxon signed rank test (p-value) |
|---|---|---|---|---|---|---|---|---|---|---|
| | Pretest | | Posttest | | | Pretest | | Posttest | | |
| | n | % | n | % | | n | % | n | % | |
| **Absorbents used during menstruation** | | | | | | | | | | |
| Disposable sanitary pad | 12 | 30.0 | 11 | 27.5 | 1.000 | 19 | 47.5 | 17 | 42.5 | 1.000 |
| Homemade reusable cloth | 0 | 0.0 | 0 | 0.0 | | 4 | 10.0 | 0 | 0.0 | |
| Both disposable sanitary pad and homemade reusable cloth | 28 | 70.0 | 29 | 72.5 | | 17 | 42.5 | 23 | 57.5 | |
| Use of soap water to clean cloth | 24 | 85.7 | 29 | 100.0 | **0.025** | 17 | 81.0 | 20 | 87.0 | 0.257 |
| Sun-drying of used cloth pad | 6 | 21.4 | 29 | 100.0 | **<0.001** | 8 | 38.1 | 7 | 30.4 | 0.655 |
| Changing absorbents in every 4–6 hours | 9 | 22.5 | 33 | 82.5 | **<0.001** | 8 | 20.0 | 10 | 25.0 | 0.317 |
| Daily bathing during menstruation | 2 | 5.0 | 25 | 62.5 | **<0.001** | 3 | 7.5 | 2 | 5.0 | 0.564 |
| Cleaning of genitals every time using toilet | 20 | 50.0 | 39 | 97.5 | **<0.001** | 22 | 55.0 | 21 | 52.5 | 0.739 |
| Use of soap water to clean genitals | 11 | 27.5 | 32 | 80.0 | **<0.001** | 11 | 27.5 | 15 | 37.5 | 0.157 |
| Disposal of absorbents by burying, burning, wrapping and disposing in dustbin | 23 | 57.5 | 40 | 100.0 | **<0.001** | 17 | 42.5 | 16 | 40.0 | 0.739 |
| **Restriction followed during menstruation** | 27 | 67.5 | 20 | 50.0 | **0.008** | 29 | 72.5 | 26 | 65.0 | 0.180 |
| Do not go to temple | 25 | 62.5 | 17 | 42.5 | 0.083 | 26 | 65.0 | 25 | 62.5 | 0.317 |
| Do not carry out routine household work | 10 | 25.0 | 5 | 12.5 | 0.593 | 10 | 25.0 | 8 | 20.0 | 0.655 |
| Not attend school or play | 5 | 12.5 | 0 | 0.0 | 0.527 | 0 | 0.0 | 1 | 2.5 | 0.102 |
| Restriction on certain types of food | 1 | 2.5 | 0 | 0.0 | **0.034** | 3 | 7.5 | 2 | 5 | 0.480 |
| Don't touch males | 7 | 17.5 | 0 | 0.0 | 1.000 | 4 | 10 | 4 | 10 | 0.257 |
| Sleep separately | 10 | 25.0 | 1 | 2.5 | 0.593 | 7 | 17.5 | 6 | 15.0 | 0.480 |
| **Practice level on menstrual hygiene** | | | | | | | | | | |
| Poor practice (0–4) | 11 | 27.5 | - | - | **<0.001** | 6 | 15.0 | 8 | 20.0 | 0.116 |
| Fair practice (5–8) | 20 | 50.0 | 13 | 32.5 | | 26 | 65.0 | 23 | 57.5 | |
| Good practice (≥9) | 9 | 22.5 | 27 | 67.5 | | 8 | 20.0 | 9 | 22.5 | |

Regarding proper discard of used absorbents that is rolling the pad, put into the disposal bag, and throwing it in a dustbin only 15.0% followed this practice in the study group however after intervention, 100.0% of participants practiced proper discarding of absorbents which was made confirmed through observation of toilet's bin.

In the non-intervention group, only 10.0% discarded the used absorbents correctly in the pretest which changed to 12.5% in the posttest. On the other hand in self-reported practice-related questions, 40.0% in the non-intervention group reported discarding used absorbents after wrapping, however, in actual observation of the school's toilet bins only a few were wrapping and disposing used absorbents. This proves that self-reported responses could be biased and may not reflect actual behavior. In the intervention group, only 52.5% practiced washing their hand with soap water while in the post-test 100.0% wash hands with soap water. In the non-intervention group, less than half (40.0%) washed their hands with soap water after changing absorbents which increased slightly to 42.5% in the post-test. There was a significant statistical difference (p<0.05) in the intervention group in all observation checklists except in the selection of absorbents while no significant difference was seen in all of the observation checklists in the non-intervention group. Overall, there was significant improvement (p<0.005) in good hygienic practice which was observed through the observation checklist from 45.0% to 100.0% in the study group, while a minor difference was observed in the non-intervention group (p>0.05) which was from 25.0% to 27.5% (Table 6).

**Table 6. Observed menstrual hygiene practices in the intervention and non-intervention group.**

| Variable | Intervention group (n = 40) | | | | Wilcoxon signed rank test (pvalue) | Non-intervention group (n = 40) | | | | Wilcoxon signed rank test(p-value) |
|---|---|---|---|---|---|---|---|---|---|---|
| **Observation Checklist** | **Pretest** | | **Posttest** | | | **Pretest** | | **Posttest** | | |
| | **n** | **(%)** | **n** | **(%)** | | **n** | **(%)** | **n** | **(%)** | |
| Appropriate selection of absorbents (all types absorbents was made available in front of them) | 40 | 100.0 | 40 | 100.0 | 1.000 | 40 | 100.0 | 40 | 100.0 | 1.000 |
| Wash hand with soap water before changing absorbent (on dummy) | 0 | 0.0 | 27 | 67.5 | **<0.001** | 1 | 2.5 | 0 | 0.0 | 0.317 |
| Proper placement of menstrual product (on dummy) follow these steps; | 21 | 52.5 | 36 | 90.0 | **<0.001** | 16 | 40.0 | 18 | 45.0 | 0.317 |
| • Remove paper from back of pad | | | | | | | | | | |
| • Stick on panty and press | | | | | | | | | | |
| • Remove paper from wings pads, fold wings pad around panty and press | | | | | | | | | | |
| Proper discard of used absorbent (Roll the pad, put into the disposal bag and throw it in a dustbin) | 6 | 15.0 | 40 | 100.0 | **<0.001** | 4 | 10.0 | 5 | 12.5 | 0.564 |
| Wash hand with soap water after changing absorbent (on dummy) | 21 | 52.5 | 40 | 100.0 | **<0.001** | 16 | 40.0 | 17 | 42.5 | 0.819 |
| **Observed practice level on menstrual hygiene** | | | | | | | | | | |
| Poor observed practice ≤2.5) | 22 | 55.0 | - | - | **<0.001** | 30 | 75.0 | 29 | 72.5 | 0.572 |
| Good observed practice(>2.5) | 18 | 45.0 | 40 | 100.0 | | 10 | 25.0 | 11 | 27.5 | |

## Discussion

This paper aims to assess the effects of health education intervention on improving knowledge and practices regarding menstrual hygiene among school-going adolescent girls. The findings of this study was supported by previous papers from India and Egypt [8,27,28,35]. The study findings from India and Iran revealed statistically significant differences in menstrual hygiene between the intervention and non-intervention groups and focused on the importance of menstrual health education [8,29,36].

In this study, the median age of menarche among school-going adolescent girls was 12 years in intervention and non-intervention groups which was consistent with the previous study [36–38]. While studies carried out in Jumla district, Nepal and India [5,32] showed the age of menarche to be (13.2 (±0.1), (13.62± 0.91) years. This may be due to the influence of heredity, lifestyles, socioeconomic and nutritional status.

67.5% of participants from the intervention and 47.5% of participants from the non-intervention group were aware of menstruation before the menarche which was similar to the study conducted in India [8] while lower than the study done in Nigeria [39] and higher than a study conducted in India [40]. This variation may stem from the fact that menstruation is still often viewed as a taboo or sensitive subject, leading to a lack of accurate, comprehensive and complete information on the topic. The hesitancy to discuss menstruation openly can contribute to misinformation and gaps in knowledge. Educating every girl child about menstruation and menstrual hygienic practices before menarche is crucial because it will not only enable them to manage menstruation confidently and hygienically but it will also prepare them to pass this critical knowledge on to future generations, ensuring that more young girls are informed and ready.

The main source of information about menstruation before the menarche were mothers (82.5%) in the intervention and (90.0%) in the non-intervention group. This result was similar to the studies conducted in India, Saudi Arabia and Nepal [41–43] while studies conducted in

Ethiopia and Egypt showed that friends and media were the main sources of information about menstruation [17,44].

Despite a higher proportion of girls receiving advice from their mothers about menstruation, many still reported poor knowledge and practices. This suggests that the mothers themselves lack accurate knowledge and proper practices and the same has been transferred to the offspring. It is crucial to educate mothers to break this cycle and ensure they have the necessary information to pass on to their daughters.

A school-based cross-sectional study in western Ethiopia and a systematic review and meta-analysis conducted in India showed the limited role of the teacher in delivering menstrual hygiene education at school [17,45]. In this study, only 7.5% of participants in the intervention and 5.0% in the non-intervention group have reported gaining menstrual-related information from the teacher which shows multiple weaknesses in the existing menstrual education curriculum. The strong biological focus of school texts left little scope for meaningful discussion on constructive menstrual hygienic practice [46].

A significant difference between the pretest and post-test high knowledge scores can be observed regarding menstrual hygiene in the intervention group that is from 10.0% to 67.5%, which was similar to the study conducted in Bangladesh and Sudan [31,47] however higher than the study conducted in Jumla, Nepal [32]. These similarities and discrepancies could be attributed to the difference in research area, sample size and study design.

According to this study and other similar studies, there have been substantial improvements in knowledge about menstruation and menstrual hygiene after intervention($p<0.001$) in items like menstruation as a normal phenomenon, hormone being the cause of menstruation, participants being aware of normal age to attain menarche, menstrual blood being not impure and poor menstrual hygiene leading to infection [10,32,41,47–49]. While another study conducted in a rural area of Haryana, negligible increment in knowledge regarding items like organ for menstrual blood flows and menstrual blood being pure was reported this may be due to the family's poor socioeconomic status and low education level given that the study was conducted in a rural part of India [40].

Regarding hygienic practice during menstruation, this study illustrates statistically significant differences in the intervention group before and after the program in all items with p-value $<0.001$ which was similar to the studies conducted in India, Egypt, Saudi and Bangladesh [5,10,20,31,49,50].

In the intervention group, only 30.0% in the pretest and even lower 27.5% in the posttest girls reported using sanitary pads during the menstruation period which was lower than studies conducted in India and Chitwan, Nepal [5,43,48], it may be because of the high cost associated with a sanitary pad and economic status of the family where the majority of parent's occupation was daily wages.

In intervention group, sun drying of used cloth pads after the intervention was high (21.4% vs 100%) which was parallel with previous studies [5,31] and higher than a study conducted in Jumla, Nepal [32]. The study shows an increase in the frequency of changing absorbents four times or more in the posttest which is 82.5% which was higher than the study conducted in India [48] and lower than the study conducted in Egypt [10] this difference could be due to different study settings.

Regarding daily bathing during menstruation, a significant difference can be observed in the intervention group from 5.0% to 62.5% ($p<0.05$) in comparison to the non-intervention group however, it was lower than the study conducted in Sudan [47] and higher than the study conducted in Jumla, Nepal [32]. This variation may be due to a lack of knowledge regarding bathing during the menstruation period.

The significant result can be seen in practice regarding disposal of absorbents in the dustbin after wrapping from pretest to posttest that is (57.7% vs 100.0%) which was greater than the study conducted in India [48], which might be due to the use of simulation approach during intervention where participants were involved themselves for safe disposal of absorbents.

Previous studies carried out in Egypt, India and South Africa showed that girls wrapped used pads in paper or plastic before discarding them [13,51,52] and this method is being encouraged by some absorbent manufacturing companies or has been included in solid waste segregation measures [53]. Even though incinerators are a better method to discard menstrual waste they have been reported to improve school sanitation facilities as a whole by 42.0% and make it easy for girls to change pads in school by 34.0% [54]. Limited papers on the use of incinerators in low and middle-income countries(LMICs) including Nepal have been reported due to budget, capacity for operation and maintenance and also limited research has been conducted on this issue [53]. The provision of services such as a handwashing site and availability of soap water including menstrual hygiene materials are required to promote sustainable menstrual hygiene practices among school-going adolescent girls in Nepal, which is consistent with the latest articles published in the International Journal of Environmental Research and Public Health [55].

Before the intervention, 62.5%, 12.5%, 17.5%, and 25.0% of girls reported facing restrictions during menstruation, such as not visiting temples, not attending school or playing, not touching males, and sleeping separately respectively. No significant differences were observed regarding those restrictions after intervention (p>0.05) which were also similar to other studies conducted in Bangladesh and Nepal [31,32]. Inclusion of comprehensive menstrual hygiene education in the school curriculum before the menarche and engagement of mothers, teachers, school nurses and community leaders in menstrual hygiene education sessions can be more effective in improving menstrual hygiene in the long run which will allow girls to discuss freely about menstrual issues and minimize harmful traditional practices.

The main strength of this study lies in its well-planned educational intervention, which successfully improved menstruation-related knowledge and practices among the participants. The simulation approach focused on hygienic menstrual practices like washing hands, appropriate selection of absorbents and safe disposal allowing participants to observe and get involved in those hygienic activities in a real setting by themselves. There has been few intervention research on improving basic school girls' menstrual hygiene knowledge and practices. Despite the fact the age of menarche has been continuously dropping in recent years, many primary or basic schools and communities remain unprepared. The limitation of this study was that the majority of knowledge and practice-related data were self-reported as a result, it was assumed that participants understood the questions and that their responses reflected their real-life behavior may not have been accurate. Another limitation of this study was the small sample size, as only two schools were selected. This may compromise the generalizability of the results.

## Conclusions

Before the intervention, adolescent girls had a limited understanding of menstruation and proper menstrual hygiene practices. However, after the health education sessions, their knowledge and hygienic practices significantly improved. This study highlights the importance of conducting comprehensive and practical menstrual hygiene educational intervention in primary or basic schools aiming at early teenage girls considering timing, content and method of delivery. Prioritizing girl-friendly WASH facilities is paramount as it not only enhances their overall menstrual health and quality of life but also plays a crucial role in improving their

education through confident management of menstrual hygiene without facing any barriers or societal stigma and discrimination.

## Supporting information

**S1 Dataset.**
(SAV)

## Acknowledgments

The authors extend heartfelt thanks to all participants and teachers from their respective schools for their support and cooperation, without whom this research endeavor would not have been possible. The authors also expresses gratitude to the Health Division of Pokhara Metropolitan City for funding the study and provider of the educational materials whose assistance was crucial to the intervention's success.

## Author Contributions

**Conceptualization:** Saraswati Ghimire.

**Data curation:** Saraswati Ghimire, Niranjan Shrestha, Smriti Manandhar.

**Formal analysis:** Saraswati Ghimire, Niranjan Shrestha, Shalik Ram Dhital.

**Funding acquisition:** Saraswati Ghimire.

**Investigation:** Saraswati Ghimire.

**Methodology:** Saraswati Ghimire, Nand Ram Gahatraj, Shalik Ram Dhital.

**Project administration:** Saraswati Ghimire.

**Resources:** Saraswati Ghimire, Smriti Manandhar, Shalik Ram Dhital.

**Software:** Nand Ram Gahatraj, Niranjan Shrestha.

**Validation:** Nand Ram Gahatraj.

**Writing – original draft:** Saraswati Ghimire.

**Writing – review & editing:** Saraswati Ghimire.

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
