## [Decision Letter · Decision Letter 0]

26 Mar 2024

PONE-D-23-28751Effects of health education intervention on menstrual hygiene knowledge and practices among the adolescent girls of Pokhara Metropolitan NepalPLOS ONE

Dear Dr. Ghimire,

Thank you for submitting your manuscript to PLOS ONE. After careful consideration, we feel that it has merit but does not fully meet PLOS ONE’s publication criteria as it currently stands. Therefore, we invite you to submit a revised version of the manuscript that addresses the points raised during the review process. We would like to see the writing part more clear as recommended by the reviewers. Please revise accordingly.

We look forward to receiving your revised manuscript.

Kind regards,

Khem Narayan Pokhrel, Ph.D.

Academic Editor

PLOS ONE

A clean copy of the edited manuscript (uploaded as the new *manuscript* file)”.

6. We note that all Figure in the supporting information in your submission contain copyrighted images. All PLOS content is published under the Creative Commons Attribution License (CC BY 4.0), which means that the manuscript, images, and Supporting Information files will be freely available online, and any third party is permitted to access, download, copy, distribute, and use these materials in any way, even commercially, with proper attribution. For more information, see our copyright guidelines: http://journals.plos.org/plosone/s/licenses-and-copyright.

1. You may seek permission from the original copyright holder of all Figure in the supporting information to publish the content specifically under the CC BY 4.0 license. 

Reviewers' comments:

Reviewer's Responses to Questions

**Comments to the Author**

1. Is the manuscript technically sound, and do the data support the conclusions?

Reviewer #1: Partly

Reviewer #2: No

2. Has the statistical analysis been performed appropriately and rigorously? 

Reviewer #1: Yes

Reviewer #2: No

3. Have the authors made all data underlying the findings in their manuscript fully available?

Reviewer #1: Yes

Reviewer #2: Yes

4. Is the manuscript presented in an intelligible fashion and written in standard English?

Reviewer #1: No

Reviewer #2: No

5. Review Comments to the Author

Reviewer #1: Research topic is very relevant and one of the public health problem of Nepal.There are some comments to the author:

1. Abstract should be limited to the 300 words, please follow the author's guideline of the journal.

2. It is better to put a keywods in alphabetical order.

3. Please include the line number in manuscript file, it would be easier to comment and author would easily addressed those comments.

4. Please complete the sentence appropriately, some of the lines are difficult to understand and some are incomplete.

5. While citing in the text, please include the author's name or where the study is done instead of writing most of the studies, some studies.

6. Would you please justify choosing the study site, as only government schools were choosen and only two schools were taken.

7. Would you please give a reference regarding the age of adolescents.

8. Have you gave any educational sessions to the non-intervention group after the study survey? As schools were randomly assigned as interventional and non-interventional groups.

9. It is very unclear that why the participants having irregular periods were excluded from the study, would you like to give a clarification regarding this?

10. Would you like to explain this sentence, 'A systematic review of existing scientific articles, official reports and policies on menstrual hygiene was done' and how it was done?

11. Would you please mention the version of SPSS and license number as data was tranformed into SPSS.

12. Would you please mention the reference of categorization of observed practice level.

13. In table number 3,4 and,5 statistical methods were not mentioned.

14. It would be better if you could shows the associations between knowledge and practice of the participants.

15. As the sample size is very small, results could not be generalised to overall Pokhara municipality.

16. In discussion, how you compare with median age of the study with average age of participants of other studies?

17. Would you please write the conclusions of the study rather than writing the recommendations.

Reviewer #2: Effects of health education intervention on menstrual hygiene knowledge and practices among the adolescent girls of Pokhara Metropolitan Nepal

Summary

Firstly, the authors must be congratulated for their efforts to shed light on a major public health issue in LICs and LMICs. Menstrual hygiene and practices in any community are a clear reflection of the surrounding socio-cultural dimensions. Adolescent girls are an important intervention group for improving the situation. The present study has rightly focused on this group in two government schools in the Pokhara region of Nepal. Their intervention has shown significant improvement in the menstrual knowledge and practices of the participants.

Comments

1. The language of the manuscript is poor and will require extensive improvements to make it readable. Repeated words, poor grammar, and conflicted sentence structure make it a difficult read.

2. The Abstract is lengthy and the Methods do not describe the intervention adequately. Major restructuring is required. Consider this sentence as a reflection for the language in the entire manuscript – “A pre-test was done using structured questionnaire and observation checklist for data collection on menstrual hygiene followed by health education interventional sessions in the intervention groups and no any session were conducted in non-intervention group.”

3. The sample size of 80 makes it a pilot study. The tools utilized were piloted in 10% of the total sample (n=8). Difficult to make statements about validity and reliability with 8 participants.

4. Not giving any intervention to the control group despite knowing the problem there, raises ethical concerns. Educational intervention in one group, educational + practical + discussion in another group would have been ethically suited. Have the authors at least provided any intervention to the control group after the end of the study?

5. Only a single 2 hr 10 min intervention in one month for the group seems inadequate. What have the authors done to avoid socially desirable responses?

6. It is recommended that the authors concise the manuscript to about 1000 words (200 Intro, 250 Methods, 250 Results and 300 Discussion) and submit it to another journal as a Correspondence / Letter to Editor / Brief Communication.

7. With this, it is not possible to recommend this manuscript in its current form for publication.

6. PLOS authors have the option to publish the peer review history of their article (what does this mean?). If published, this will include your full peer review and any attached files.

Reviewer #1: No

Reviewer #2: No

---

## [Author Response · Author response to Decision Letter 0]

9 May 2024

Dear Academic Editor and Reviewer,

We are honored and delighted to have been given the opportunity for submitting a revised version of the manuscript that addresses the points raised during the review process.

In face of PLOS ONE's decision for the manuscript ("Decision: Revision required [PONE-D-23-28751]-[EMID: b2f3fa8ec33e93d6]. We appreciate the insightful feedback provided by the editor and reviewers, which helped us to refine the revised version of the manuscript that was submitted.

All the comments were considered providing the following changes to the points raised by the academic editor and reviewers. 

The name of the colleague or the details of the professional service that edited your manuscript:

Response: The identity of the associate that proofread this manuscripts is mentioned. 

A copy of your manuscript showing your changes by either highlighting them or using track changes (uploaded as a *supporting information* file).

Response: A copy of manuscript showing changes was done using track changes and it is uploaded as a *supporting information* file.

A clean copy of the edited manuscript (uploaded as the new *manuscript* file)”.

Response: A clean copy of the edited manuscript has been uploaded as the new *manuscript* file)”.

Note that the grant information you provided in the ‘Funding Information’ and ‘Financial Disclosure’ sections do not match. 

Response: My apologies, it has been now been corrected. Thank you 

Ensure that to provide the correct grant numbers for the awards you received for your study in the ‘Funding Information’ section.

Response: Apologies for the oversight. The grant number was 438 with amount, NPR 40,000 (USD 300), which was very small amount which was only used for the data collection.

Your ethics statement should only appear in the methods section of your manuscript. If your ethics statement is written in any section besides the Methods, please move it to the Methods section and delete it from any other section.

Response: Thank you for your note. An ethical statement is now written in the end of the methods section.

All Figure in the supporting information in your submission contain copyrighted images. We require you to either (1) present written permission from the copyright holder to publish these figures specifically under the CC BY 4.0 license, or (2) remove the figures from your submission:

Response: Thank you for asking a copy right permission letter. Unfortunately we do not have written permission of images. Therefore, we would like to remove all the figures if it is unethical. However, majority of photos were taken from own mobile phone. Therefore we have decided to remove those images. 

Reviewer’s comments:

Reviewer #1

1. Abstract should be limited to the 300 words, please follow the author's guideline of the journal.

Response: Thank you for your comments. The revised manuscript abstract is limited to the required words based on journal author's guideline. Now the abstract word count is 300 excluding keywords.

2. It is better to put a keywords in alphabetical order.

Response: Thank you for your guidance. Keywords are now arranged in alphabetical order in the revised manuscript.

3. Please include the line number in manuscript file, it would be easier to comment and author would easily addressed those comments.

Response: Thank you for your suggestion. We have now created line numbers in the revised manuscript. 

4. Please complete the sentence appropriately, some of the lines are difficult to understand and some are incomplete.

Response: Thank you for your valuable feedback. All the sentences are checked appropriately and completed in understandable language in revised manuscript. 

5. While citing in the text, please include the author's name or where the study is done instead of writing most of the studies, some studies.

Response: Thank you for making us more sincere. Now, we have added original citation in the text and the study location is now included instead of just writing most of the studies or some studies. 

6. Would you please justify choosing the study site, as only government schools were chosen and only two schools were taken.

Response: We appreciate your question. Thank you. We would like to give some insight on it. The previous studies conducted at schools especially among South Asian and Sub-Saharan African showed, girls who attend government schools were more likely to have poor menstrual hygiene because of their low level of socioeconomic status [19].The selection of government schools as the study site was deliberate and aimed to ensure a representative sample of adolescent girls from a specific socioeconomic background who might be most in need of support. Government schools were chosen for their accessibility and cooperation in research activities. Additionally, two schools were selected for in-depth investigation within the constraints of available resources, time and logistical challenges associated with executing the intervention. (Reference is cited in manuscript).

7. Would you please give a reference regarding the age of adolescents.

Response: Adolescent’s age group age between 10-15 is crucial period where adolescents girls had to deal with fear, stigma, discomfort, superstition and many more during menstruation especially in poor countries like Nepal. In this period right information and practice should be promoted so that their menstrual health of those adolescents is maintained for long term. The research conducted on Nepal, Egypt, Kenya, South Africa, India have recommended for conducting menstrual hygiene and health interventions as early as possible so as to equip adolescent’s girls with scientific knowledge and hygienic practices regarding menstruation [16,21–25]. Average age of menarche among school going adolescents was between 10-15 years in this study and study conducted by these studies [5,28, 35–37]. Even though national school curriculum on menstrual hygiene in textbooks have been updated from grade 4-12 [11], still more than half adolescents have inadequate information about menstruation and only one in ten are practicing good menstrual hygiene[12].The research conducted in developing nations such as Nepal, India, and Nigeria [5, 28, 35–37] revealed that the mean age of menarche among school-going adolescents falls within the range of 10 to 15 years. Research conducted by Fetohy EM in Egypt and Chang and Chen in the Hualien Region has highlighted the importance of expanding menstrual hygiene education to elementary, preparatory, and other secondary schools, especially during crucial transitional periods such as menarche. Furthermore, their studies have demonstrated the effectiveness of educational initiatives in these population in enhancing menstrual health [21,27]. Therefore, this age group was selected as the target population for the study. (Reference is cited in manuscript).

8. Have you gave any educational sessions to the non-intervention group after the study survey? As schools were randomly assigned as interventional and non-interventional groups.

Response: Yes. We have conducted educational sessions like video playing, PowerPoint presentation, group discussion, simulation, handouts distribution in non-intervention group as well after the study survey.

9. It is very unclear that why the participants having irregular periods were excluded from the study, would you like to give a clarification regarding this?

Response: Thank you for raising this issue. Participants having irregular periods were excluded from the study especially in data collection period but not while conducting sessions. It was done to ensure a more homogeneous sample, allowing for a clearer assessment of the intervention's effects on menstrual practices among individuals with regular cycle. Given the short duration of the intervention (one month), participants with irregular menstrual cycles may not experience the full benefits of the intervention or may have difficulty adhering to the study practical aspect which was done through simulation.

10. Would you like to explain this sentence, 'A systematic review of existing scientific articles, official reports and policies on menstrual hygiene was done' and how it was done?

Response: Thank you for the note. We apologize, it was intended to say that performing a literature review prior to conducting a study, but the way it was written conveyed an inaccurate message, thus it has been fixed now in the revised manuscript.

11. Would you please mention the version of SPSS and license number as data was transformed into SPSS.

Response: SPSS version 21 was used. We have included this information in our revised manuscript. Thank you.

12. Would you please mention the reference of categorization of observed practice level?

Response: Our literature review did not find any published article that provided guidance on categorizing ‘observed practice level’. Therefore, we used median as cut off point to categorize as poor and good practice as data were not normally distributed. If it is unsuitable to categorize observed practice level this way it can be discarded. 

13. In table number 3, 4 and, 5 statistical methods were not mentioned.

Response: Statistical methods now has been mentioned in table number 3, 4 and 5 and below table. We have also included this information in statistical section and have indicated respective table number. In table 3, Chi square and Fisher’s exact test was used to assess difference between interventional and non-interventional group regarding socio-demographic information. For inferential statistics Wilcoxon signed rank test was used.

14. It would be better if you could shows the associations between knowledge and practice of the participants.

Response: We appreciate your question. Thank You for insight. The primary aim of this paper was to assess the effects of a health education intervention on menstrual hygiene knowledge and practices among adolescent girls, rather than to explore the associations. Because sample sizes are small, it may lack the ability to detect true associations between groups and may not provide meaningful or reliable results. We will consider examining the association between knowledge and practice of the participant’s in future interventional studies with larger sample sizes.

15. As the sample size is very small, results could not be generalized to overall Pokhara municipality.

Response: Thank you for this feedback. We have included this as a limitation of our study. 

16. In discussion, how you compare with median age of the study with average age of participants of other studies? 

Response: In my study, the data was not normal. So, non-parametric test was used and median age was used. In similar kind of studies, the data was parametric and hence, mean age was used. As, my age categorization was very narrow between 10-15 years there was not much difference in age comparison with other study that have used mean. As aim of the study was to assess the effects of health education intervention on menstrual hygiene knowledge and practice on menstrual hygiene. Age comparison could be given less important if needed based on your feedback comparison with average will be discarded.

17. Would you please write the conclusions of the study rather than writing the recommendations?

Response: Thank you for your valuable comment. Conclusions of the study is now revised based on your suggestions. Now it has been written with key findings of the study. 

Reviewer #2: 

1. The language of the manuscript is poor and will require extensive improvements to make it readable. Repeated words, poor grammar, and conflicted sentence structure make it a difficult read.

Response: Thank you for your insightful comment. The language review is now done by an academic and professional reviewer. We revised manuscript thoroughly and rechecked for repeated words, poor grammar and conflicted sentence structure.

2. The Abstract is lengthy and the Methods do not describe the intervention adequately. Major restructuring is required. Consider this sentence as a reflection for the language in the entire manuscript – “A pre-test was done using structured questionnaire and observation checklist for data collection on menstrual hygiene followed by health education interventional sessions in the intervention groups and no any session were conducted in non-intervention group.”

Response: Thank you for the comment. Due to constraints on word count, the methods section limited detailed description of the intervention. Therefore, detail information regarding the intervention has been included in the materials and methods section. Now the abstract is limited to 300 words in consistency with first reviewer’s comment. 

3. The sample size of 80 makes it a pilot study. The tools utilized were piloted in 10% of the total sample (n=8). Difficult to make statements about validity and reliability with 8 participants.

Response: Thank you for the insightful question. Although our health education intervention results indicate promising outcomes for the studied population, it's crucial to acknowledge the limits of our research, especially the small sample size. Piloting tools on only 10% of the sample (n=8) may indeed limit the ability to make definitive statements about validity and reliability. It's important to acknowledge this limitation and consider it when interpreting the results. A pilot study was conducted at one basic school, involving the distribution of questionnaires. The questionnaire were pretested to check the reliability and validity. The aim was to assess the feasibility of the planned research and identify any issues with the questionnaire. Few corrections came from the pilot study, and later all questions were found to be clear. 

4. Not giving any intervention to the control group despite knowing the problem there, raises ethical concerns. Educational intervention in one group, educational + practical + discussion in another group would have been ethically suited. Have the authors at least provided any intervention to the control group after the end of the study?

Response: Thank you for your comments. Health education session was conducted in control group after posttest in both group using PowerPoint, video, simulation, handouts distribution, group discussion knowing the presence of menstrual hygiene issues in non-intervention group as well.

5. Only a single 2 hr 10 min intervention in one month for the group seems inadequate. What have the authors done to avoid socially desirable responses?

Response: We are grateful for your concerns. We focused on using resources efficiently, involving PowerPoint presentations, videos, group discussions and simulations which were more focused on practical aspect that convey powerful message. This approach aimed to improve both understanding and practice of menstrual hygiene within short period of time, avoiding the need for lengthy and repetitive sessions. Designing of intervention was drafted by first author, then discussion was done with expert and supervisors then was finalized with the help of supervisors. Data were collected by fourth author and first authors facilitated in collection who were taken training from the university prior data collection. Concurrent supervision was done by the supervisors. All data were kept securely and privacy was maintain in the password protected laptop. 

Author have carried out following activity to avoid socially desirable responses

• Person who gave the intervention was different from one who administered the questionnaire and was given once because participants in repetitive educational interventions may feel pressure to conform to perceived social norms or expectations, leading to social desirability bias. They may provide responses that they believe are socially acceptable or desirable, rather than expressing their genuine opinions or behaviors leading to biased conclusions. .

• It was anonymous surveys to encourage participants to provide honest responses without fear of judgment or societal expectations.

6. .It is recommended that the authors concise the manuscript to about 1000 words (200 Intro, 250 Methods, 25

---

## [Decision Letter · Decision Letter 1]

2 Jul 2024

PONE-D-23-28751R1Effects of health education intervention on menstrual hygiene knowledge and practices among the adolescent girls of Pokhara Metropolitan NepalPLOS ONE

Dear Dr. Ghimire,

Thank you for submitting your manuscript to PLOS ONE. After careful consideration, we feel that it has merit but does not fully meet PLOS ONE’s publication criteria as it currently stands. Therefore, we invite you to submit a revised version of the manuscript that addresses the points raised during the review process.

The manuscript has been evaluated by two reviewers, and their comments are available below.

One of the reviewers has a few remaining minor concerns. Could you please carefully revise the manuscript to address all comments raised?==============================

We look forward to receiving your revised manuscript.

Kind regards,

Johanna Pruller, Ph.D.

Associate Editor

PLOS ONE

Journal Requirements:

Reviewers' comments:

Reviewer's Responses to Questions

**Comments to the Author**

1. If the authors have adequately addressed your comments raised in a previous round of review and you feel that this manuscript is now acceptable for publication, you may indicate that here to bypass the “Comments to the Author” section, enter your conflict of interest statement in the “Confidential to Editor” section, and submit your "Accept" recommendation.

Reviewer #1: All comments have been addressed

Reviewer #2: (No Response)

2. Is the manuscript technically sound, and do the data support the conclusions?

Reviewer #1: Partly

Reviewer #2: Yes

3. Has the statistical analysis been performed appropriately and rigorously? 

Reviewer #1: Yes

Reviewer #2: Yes

4. Have the authors made all data underlying the findings in their manuscript fully available?

Reviewer #1: Yes

Reviewer #2: Yes

5. Is the manuscript presented in an intelligible fashion and written in standard English?

Reviewer #1: No

Reviewer #2: Yes

6. Review Comments to the Author

Reviewer #1: Thank you for your response to the comments. It is really appreciable for your hard work and efforts. The manuscript is a bit lengthy so you can summarize a results in brief.

Reviewer #2: The authors have done a good job in revising the manuscript. It is a relevant topic and has the potential to be replicated in different socio-economically similar settings.

The previous comments have been addressed satisfactorily. I have few minor questions / suggestions -

1. Authors have described how the schools were selected. A justification on how the participants were selected in the classrooms is required. How was the sample size achieved? What did the authors do if the total number of eligible participants went beyond 40? Were some participants rejected just because the sample size of 40 was achieved? Why didn't the authors think of having similar proportion of participants from class 6 to 8?

2. There is no mention of Assent form. Generally for participants aged 12-18 years, an assent from participant and consent from parents is required. Was a written informed assent taken?

3. There are 6 tables in the Results Section. Plus there is a lot of text. The authors should revise this section to avoid / shorten repeated information present in the both the tables and text.

4. Who administered the intervention? What training did that individual receive? Was any pre-intervention rapport built with the participants?

5. As mentioned earlier, the study is relevant. The authors should consider a follow up study of the same group of participants to assess their knowledge and practices at 6 months and 1 year post-intervention to gauze the compliance of the participants to the menstrual hygiene methods included in the intervention.

7. PLOS authors have the option to publish the peer review history of their article (what does this mean?). If published, this will include your full peer review and any attached files.

Reviewer #1: No

Reviewer #2: No

---

## [Author Response · Author response to Decision Letter 1]

15 Aug 2024

Dear Academic Editor and Reviewer,

We are honored and delighted to have been given the opportunity to submit final revised version of the manuscript that addresses the points raised during the review process.

In the face of PLOS ONE's decision for the manuscript ("Decision: Revision required [PONE-D-23-28751R1]-[EMID: c7e1a1d379c8c8d6]. We appreciate the insightful feedback provided by the editor and reviewers, which helped us refine the revised version of the submitted manuscript.

All the comments were considered providing the following changes to the points raised by the academic editor and reviewers. 

Upon resubmission, we have provided the following:

Response: A rebuttal letter that responds to each point raised by the academic editor and reviewer(s) has been uploaded as a separate file labeled 'Response to Reviewers'.

Response: A marked-up copy of our manuscript that highlights changes made to the original version has been uploaded as a separate file labeled 'Revised Manuscript with Track Changes'.

Response: An unmarked version of our revised paper without tracked changes. This has been uploaded as a separate file labeled 'Manuscript'.

Regarding the financial disclosure, no changes are required at this time it was a very small amount and was utilized only for data collection.

Journal Requirements:

Response: All the reference lists were reviewed to ensure completeness and correctness. The cited papers that were retracted for example reference list number 18 and 53 were removed. Reference list numbers 6,10,15,16,20,22,24,30,37,40,41,51, was checked for completeness. Reference lists numbers 6 and 15 were made as one after checking for completeness and correctness as both references were from the same paper.

Reviewer’s comments:

1. Authors have described how the schools were selected. A justification on how the participants were selected in the classrooms is required. How was the sample size achieved? What did the authors do if the total number of eligible participants went beyond 40? Were some participants rejected just because the sample size of 40 was achieved? Why didn't the authors think of having a similar proportion of participants from class 6 to 8?

Response: Thank you for the insights. The sample size of 40 participants was achieved based on the previous study which we have cited in the materials and methods section (sample size calculation part) in the manuscript. Participants were not rejected simply to achieve a sample size of 40 rather every effort was made to include participants who met eligible criteria like age, attendance of menarche, obtaining written informed assent, prioritizing inclusivity and ethical considerations within the predetermined sample size. 

First of all eligible participants were listed from each grade then samples were taken proportionally using a random sampling technique through the lottery method. This approach ensured that every eligible participant had an equal chance of being selected, thus maintaining the integrity of the sampling process. While we acknowledge the value of having a similar proportion of participants from grades 6 to 8, the total number of eligible participants was not uniform across these grades. Therefore, a strict proportionate sampling was not feasible due to variations in enrollment numbers across grades within the selected schools. As long as the total sample size did not exceed the study's capacity to manage and analyze data effectively, a balanced proportion across grade levels was not mandated. A similar approach was used in the non-intervention group as well.

2. There is no mention of Assent form. Generally for participants aged 12-18 years, an assent from participant and consent from parents is required. Was a written informed assent taken?

Response: Thank you for your questions. We apologize, we forgot to mention written informed assent which was taken before a study and thus it has been corrected in the revised manuscript. Written informed assent from all participants and consent from their parents was obtained in coordination with the school head teacher, ensuring ethical standards and that all participants willingly participated in the study.

3. There are 6 tables in the Results Section. Plus there is a lot of text. The authors should revise this section to avoid / shorten repeated information present in the both the tables and text.

Response: We appreciate the reviewer's comments. To enhance clarity and conciseness, we have revised this section to eliminate repetitive information. We have ensured that each table is succinctly summarized in the text, avoiding unnecessary repetition of data presented in the tables to ensure that the information is presented clearly and efficiently. We have explained the most significant and key results in the sentence presentation.

4. Who administered the intervention? What training did that individual receive? Was any pre-intervention rapport built with the participants? 

Response: Thank you for your valuable comments. The first author/ principal investigator Saraswati Ghimire (SG) conducted health education intervention. SG received one-day orientation session which was provided by research supervisors on the theoretical foundations and practical aspects of the intervention, including specific methodologies, content development, ethical considerations and participant interaction techniques.

SG possesses a specialized academic background in health promotion and education. This expertise encompasses a deep understanding of effective health educational interventions, including content development, methods of delivery and suitable use of media for disseminating health-related information. Through academic coursework and practical experience, the first author had developed proficiency in implementing health education interventions effectively. Specifically, her familiarity with menstrual hygiene education includes comprehensive knowledge of relevant topics, evidence-based approaches and culturally appropriate methods for delivering educational content through literature review and supervisor guidance. In developing the menstrual hygiene education content, the first author applied their expertise in selecting relevant information and utilizing appropriate media channels. This ensured that the intervention was not only informative but also engaging and accessible to the target audience for example through simulation. The first author's background in health promotion and education played a pivotal role in shaping the intervention’s effectiveness and relevance within the study context. 

Pre-intervention rapport building with the participants was deliberately avoided to prevent potential biases or influences on their responses. By skipping rapport building, the research maintained a neutral and unbiased environment, ensuring the integrity and reliability of the data.

5. As mentioned earlier, the study is relevant. The authors should consider a follow up study of the same group of participants to assess their knowledge and practices at 6 months and 1 year post-intervention to gauze the compliance of the participants to the menstrual hygiene methods included in the intervention.

Response: Thank you for your insightful suggestion regarding a follow-up study to assess participant's knowledge and practices at 6 months and 1-year post-intervention.

The study was limited to a short period due to several potential drawbacks such as participant attrition over time could have result in a reduced and biased sample, while recall bias may cause participants to struggle with accurately remembering past behaviors. Additionally, the Hawthorne Effect might have lead participants to alter their behavior simply because they know they are being studied. External factors, such as social influence, may have impact participant's practices independently of the intervention. Long-term studies are costly and resource-heavy, demanding considerable time and money. Moreover, ethical issues like protecting participant privacy and maintaining ethical practices over time add to the challenges.

Our study focused on the immediate impact of the intervention with focused on using resources efficiently. High-impact interventions through the use of PowerPoint presentations, videos, group discussions mini-lectures, simulations and handout distribution focused on practical aspects that convey powerful messages. This approach aimed to improve both understanding and practice of good menstrual hygiene within a short period of time, avoiding the need for lengthy study. Our study findings showed significant improvement in the knowledge and practice level of adolescent girls on menstrual hygiene after health education intervention which was of short duration.

We sincerely appreciate the editors and reviewers for their valuable insights into our current submission. We have thoroughly addressed all concerns raised by both the reviewers and the editor, ensuring our revisions align with their expectations. We are optimistic that our article will be published this time, making a significant contribution to the scientific community. We aim to raise awareness of menstrual health education globally, ensuring more people are informed about this crucial topic.

---

## [Decision Letter · Decision Letter 2]

3 Sep 2024

Effects of health education intervention on menstrual hygiene knowledge and practices among the adolescent girls of Pokhara Metropolitan, Nepal

PONE-D-23-28751R2

Dear Dr. Ghimire,

We’re pleased to inform you that your manuscript has been judged scientifically suitable for publication and will be formally accepted for publication once it meets all outstanding technical requirements.

Kind regards,

Alison Parker

Academic Editor

PLOS ONE

Additional Editor Comments (optional):

Reviewers' comments:

Reviewer's Responses to Questions

**Comments to the Author**

1. If the authors have adequately addressed your comments raised in a previous round of review and you feel that this manuscript is now acceptable for publication, you may indicate that here to bypass the “Comments to the Author” section, enter your conflict of interest statement in the “Confidential to Editor” section, and submit your "Accept" recommendation.

Reviewer #1: (No Response)

Reviewer #2: All comments have been addressed

2. Is the manuscript technically sound, and do the data support the conclusions?

Reviewer #1: No

Reviewer #2: Yes

3. Has the statistical analysis been performed appropriately and rigorously? 

Reviewer #1: Yes

Reviewer #2: Yes

4. Have the authors made all data underlying the findings in their manuscript fully available?

Reviewer #1: Yes

Reviewer #2: Yes

5. Is the manuscript presented in an intelligible fashion and written in standard English?

Reviewer #1: No

Reviewer #2: Yes

6. Review Comments to the Author

Reviewer #1: Thank you for the responses of the comments and your efforts to make a changes in manuscripts but all the comments are not justifiable.

Reviewer #2: (No Response)

7. PLOS authors have the option to publish the peer review history of their article (what does this mean?). If published, this will include your full peer review and any attached files.

Reviewer #1: No

Reviewer #2: No

---

## [Editor Report · Acceptance letter]

9 Sep 2024

PONE-D-23-28751R2 

PLOS ONE

Dear Dr. Ghimire, 

I'm pleased to inform you that your manuscript has been deemed suitable for publication in PLOS ONE. Congratulations! Your manuscript is now being handed over to our production team.

Kind regards, 

on behalf of

Dr. Alison Parker 

Academic Editor

PLOS ONE